# c-FLIP facilitates ZIKV infection by mediating caspase-8/3-dependent apoptosis

**Shengze Zhang**[1,2☉], **Nina Li**[1,2,3☉], **Shu Wu**[1,2], **Ting Xie**[1,2], **Qiqi Chen**[1,2], **Jiani Wu**[1,2], **Shike Zeng**[1,2], **Lin Zhu**[1,2], **Shaohui Bai**[1,2], **Haolu Zha**[1,2], **Weijian Tian**[1,2], **Nan Wu**[4], **Xuan Zou**[5], **Shisong Fang**[5], **Chuming Luo**[1,2], **Mang Shi**[6], **Caijun Sun**[1,2,7], **Yuelong Shu**[1,7,8], **Huanle Luo**[1,2,7]*

1 School of Public Health (Shenzhen), Shenzhen Key Laboratory of Pathogenic Microbes and Biosafety, Shenzhen Campus of Sun Yat-sen University, Shenzhen, P.R. China, 2 School of Public Health (Shenzhen), Sun Yat-sen University, Guangzhou, P.R. China, 3 Chinese Academy of Medical Sciences & Peking Union Medical College, Beijing, P. R. China, 4 Shenzhen Nanshan Center for Disease Control and Prevention, Shenzhen, P.R. China, 5 Shenzhen Center for Disease Control and Prevention, Shenzhen, P.R. China, 6 The Centre for Infection and Immunity Studies, School of Medicine, Shenzhen Campus of Sun Yat-sen University, Sun Yat-sen University, Shenzhen, China, 7 Key Laboratory of Tropical Disease Control (Sun Yat-sen University), Ministry of Education, Guangzhou, P.R. China, 8 Key Laboratory of Pathogen Infection Prevention and Control (MOE), State Key Laboratory of Respiratory Health and Multimorbidity, National Institute of Pathogen Biology, Chinese Academy of Medical Sciences & Peking Union Medical College, Beijing, P. R. China

☉ These authors contributed equally to this work.
* luohle@mail.sysu.edu.cn

**Data Availability Statement:** All relevant data are within the manuscript and its Supporting information files.

## Abstract

c-FLIP functions as a dual regulator of apoptosis and inflammation, yet its implications in Zika virus (ZIKV) infection remain partially understood, especially in the context of ZIKV-induced congenital Zika syndrome (CZS) where both apoptosis and inflammation play pivotal roles. Our findings demonstrate that c-FLIP promotes ZIKV infection in placental cells and myeloid-derived macrophages, involving inflammation and caspase-8/3-mediated apoptosis. Moreover, our observations reveal that c-FLIP augments ZIKV infection in multiple tissues, including blood cell, spleen, uterus, testis, and the brain of mice. Notably, the partial deficiency of c-FLIP provides protection to embryos against ZIKV-induced CZS, accompanied by a reduction in caspase-3-mediated apoptosis. Additionally, we have found a distinctive parental effect of c-FLIP influencing ZIKV replication in fetal heads. In summary, our study reveals the critical role of c-FLIP as a positive regulator in caspase-8/3-mediated apoptosis during ZIKV infection, significantly contributing to the development of CZS.

## Author summary

Zika virus (ZIKV) infection in pregnant women can lead to the development of Congenital Zika Syndrome (CZS) in infants, resulting in complications such as microcephaly, intrauterine growth restriction (IUGR), and miscarriages. Although the mechanisms of apoptosis and inflammatory responses in ZIKV-induced CZS are not fully understood, our study investigated the role of c-FLIP, a critical regulator of apoptosis and inflammation during ZIKV infection and its associated CZS. In both human trophoblast cells and

**Funding:** This work was supported by National Natural Science Foundation of China (32000116, 32270147; Recipient L.H.L.), Shenzhen Science and Technology Program (KQTD20200820145822023, Recipient M.S.; JCYJ20190807155407443, Recipient L.H.L.; KCXFZ20211020172545006, Recipient X.Z.; ZDSYS20230626091203007, Recipient L.H.L.), High-Level Project of Medicine in Nanshan, Shenzhen; Sanming Project of Medicine in Shenzhen (SZSM202103008, Y.L.S.), Science and Technology Planning Project of Guangdong Province in China (2021B1212040017, Recipient C.J.S.). The funders had no role in study design, data collection and analysis, decision to publish, or preparation of the manuscript.

**Competing interests:** The authors have declared that no competing interests exist.

murine-derived macrophages, we observed that c-FLIP facilitated ZIKV infection by modulating caspase-8/3-mediated apoptosis. Mice deficient in c-FLIP exhibited reduced ZIKV replication and a decrease in inflammatory cytokine production. Importantly, c-FLIP deficiency demonstrated an inhibitory effect on CZS in ZIKV-infected pregnant mice. Additionally, c-FLIP displayed a parental influence on ZIKV replication in the fetal head by triggering caspase-3 activation. This research emphasizes the significance of c-FLIP in regulating caspase-8/3 and its profound impact on ZIKV-induced CZS, providing valuable insights into the role of apoptosis in ZIKV vertical transmission and fetal development.

## Introduction

Zika virus (ZIKV) is a single-stranded, positive-sense RNA virus that belongs to the Flaviviridae family. This family includes various mosquito-borne pathogens of public concern, such as Dengue virus, Japanese encephalitis virus, and West Nile virus (WNV), which can lead to clinical outcomes ranging from asymptomatic cases to severe encephalitis [1–3]. While the flavivirus has a similar genome, ZIKV exhibits characteristics of vertical transmission associated with microcephaly and birth defects in infants. Multiple studies have demonstrated that ZIKV infection in pregnant women can lead to Congenital Zika Syndrome (CZS) in infants. CZS encompasses conditions such as microcephaly in newborns, along with other congenital abnormalities, intrauterine growth restriction (IUGR), preterm birth, and pregnancy complications, including miscarriages [4–6]. Currently, over 80 countries have reported cases of ZIKV infection, with more than 30 countries reporting ZIKV cases associated with microcephaly or other neurologic disorders [7]. However, there are no approved drugs or vaccines specifically designed for the ZIKV.

Although ZIKV has been detected in the placenta [8] and in the brains of microcephalic infants [5], the specific mechanisms through which it causes CZS remains elusive. The reported possible mechanisms include: 1) Efficient infection of neural cells by the ZIKV, leading to apoptosis and resulting in neurodevelopmental defects. 2) ZIKV induces cytokines in the placenta and neural cells, causing an excessive inflammatory response and indirectly leading to fetal brain damage [9–11]. However, the critical molecular interactions and underlying mechanisms triggering CZS through the interplay of apoptosis and inflammatory responses remain unclear. Our previous studies have revealed that the host E3 ubiquitin ligase Pellino1 not only mediates inflammatory responses but also promotes cell death in human placental cells during ZIKV infection [12]. As a crucial regulatory factor in host immune responses, Pellino1 plays a role in modulating apoptosis by influencing the levels of cellular FLICE-like inhibitory protein (c-FLIP), also known as CFLAR [13]. Among the 13 distinct spliced variants of c-FLIP, three are expressed as long (c-FLIPL), short (c-FLIPS or c-FLIPR) splice proteins in human cells, playing a significant role in the regulation of death receptor-induced apoptosis and embryonic development [14,15]. Subsequent studies have identified c-FLIP as an important regulator, acting as a switch from inflammation to cell death. These findings underscore the critical role of c-FLIP in various biological processes [16]. Additionally, c-FLIP plays a significant role in viral infections. For instance, the core protein of the Human Hepatitis C virus (HCV) in HepG2 cells inhibits apoptosis triggered by caspase-8 by sustaining the levels of c-FLIP. [17]. However, in JFH-1 and Huh-7 cells, HCV promotes apoptosis by inhibiting NF-κB, leading to reduced expression of c-FLIP and other anti-apoptotic factors [18]. Similarly, the Herpes Simplex Virus enhances apoptosis in mature dendritic cells by decreasing c-FLIP

expression [19]. Beyond apoptosis, c-FLIP can enhance the production of type I interferon induced by Coxsackievirus B3 and inhibit virus replication [20]. Additionally, c-FLIP is an essential host factor for the proliferation of the Hepatitis B virus [21].

Nevertheless, the role of c-FLIP in ZIKV pathogenesis and its underlying mechanisms remain elusive. In this study, we investigated the involvement of c-FLIP during ZIKV infection using both in vivo and in vitro models. Our findings elucidate that c-FLIP actively promotes ZIKV infection, thereby contributing to CZS through caspase-8/3-mediated apoptosis.

## Results

### c-FLIP is associated with ZIKV infection in placental cells

In our previous studies, we highlighted Pellino1's role in promoting flavivirus infections, including WNV and ZIKV [12,22]. Concurrently, another study demonstrated that Pellino1 serves as a regulator of c-FLIP [13], which prompted us to investigate whether c-FLIP plays a significant role in ZIKV infection. Considering the association of congenital diseases with ZIKV infection during early pregnancy [23], we investigated c-FLIP expression in human first-trimester extravillous trophoblast cells (HTR8). Our analysis revealed a moderate increase in *c-FLIPL* mRNA levels on day 2 and a substantial increase of *c-FLIPS* mRNA levels on day 2, day 4, and day 6 (Fig 1A and 1B) in ZIKV-infected HTR8 cells, as compared to the non-infected group. Correspondingly, an enhancement in both c-FLIPL and c-FLIPS protein levels was observed on day 1 and day 2 post-infection compared to the non-infected control (Fig 1C and 1D). However, ZIKV infection did not induce a significant change in *c-FLIPR* mRNA levels (S1A Fig), and the corresponding protein remained undetectable in both ZIKV-infected

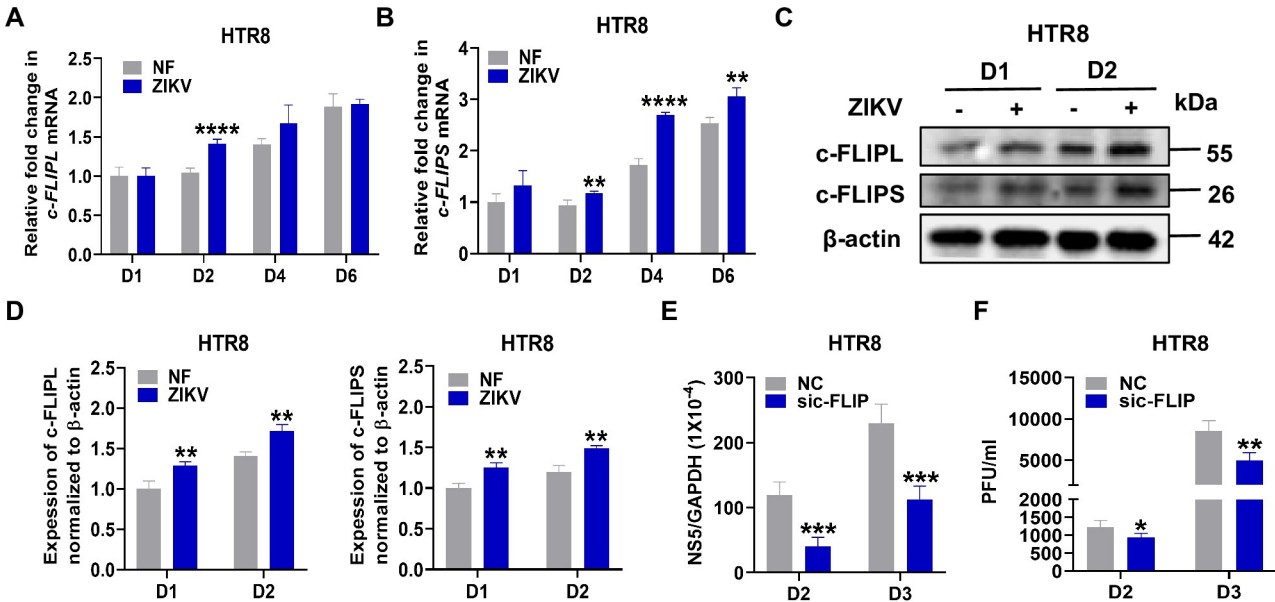

**Fig 1. c-FLIP enhances ZIKV infection in HTR8 cells.** (A-B) HTR8 cells were infected with ZIKV at a MOI of 1. c-FLIPL (A) and c-FLIPS (B) levels were measured on day 1 (D1), D2, D4 and D6 post-infection by qPCR. (C) Western blot assays of c-FLIPL and c-FLIPS expression in HTR8 cells on D1 and D2 post ZIKV infection. (D) Quantification of c-FLIPL and c-FLIPS protein levels relatives to β-actin. (E-F) HTR8 cells were infected with ZIKV at a MOI of 1 post siRNA transfection for 24 hours. The viral load was measured on D2 and D3 post-infection by qPCR (E) and plaque assay (F). The data represent either a single experiment chosen as representative from three independent experiments (A-B, E-F) or the collective results of three independent experiments (D). All the data are analyzed by unpaired Student's *t* test. Data are presented as means ± SD. *P <0.05, **P < 0.01, ***P < 0.001, ****P < 0.0001 compared to control group.

HTR8 cells and the non-infected group. These results indicate an augmentation of c-FLIPL and c-FLIPS expression in ZIKV-infected HTR8 cells. Interestingly, we observed an increase in the transcription (Fig 1A and 1B) and protein (Fig 1C and 1D) expression levels of c-FLIPL and c-FLIPS over time in the uninfected group, which we will investigate further. Nevertheless, these results indicate an augmentation of c-FLIPL and c-FLIPS expression in ZIKV-infected HTR8 cells at the indicated timepoints. Next, we executed siRNA knockdown experiments, achieving a reduction of at least 50% in the mRNA (S1B Fig) levels of c-FLIPL, c-FLIPR, and c-FLIPS, as well as the protein levels of c-FLIPL and c-FLIPS (S1C and S1D Fig) in HTR8 cells followed by infection with ZIKV. Notably, silencing c-FLIP expression in HTR8 cells using siRNA did not result in any significant effect on cell viability compared to transfection with negative control siRNA (S1E Fig). As depicted in Figs 1E, 1F and S1F, the viral loads in ZIKV-infected HTR8 cells demonstrated a 20% decrease on day 2 and a 40% decrease on day 3 post-infection compared to control siRNA-treated cells, as assessed by quantitative polymerase chain reaction (qPCR) and plaque assay. These results suggest a promotional role for c-FLIP in ZIKV infection of HTR8 cells.

## c-FLIP facilitates the replication of ZIKV in various tissues of mice

To further investigate the role of c-FLIP during ZIKV infection, we generated *c-Flip* heterozygous knockout mice using a standard CRISPR/Cas9 method, as depicted in S2A Fig. This approach was chosen because a previous study reported the non-survival of homozygous embryos by embryonic day (E) 10.5 [14]. The genotypes were verified by PCR analysis of genomic DNA (S2B Fig) and the reduced expression of c-FLIPL was confirmed at protein levels in spleen, brain, uterus, and testis (S2C–S2F Fig). Both WT and *c-Flip*$^{+/-}$ mice received pretreatment with 2mg anti-type I interferon receptor (IFNR) antibody (MAR1-5A3), followed by a challenge with $5 \times 10^5$ plaque-forming units (PFU) of the ZIKV strain one day later. Viral burdens in blood cell, spleen, brain, uterus, and testis were assessed on day 2 and day 6 post-infection. *c-Flip*$^{+/-}$ mice had significantly lower viremia on day 2 (Fig 2A and 2B) and decreased splenic viral loads on day 2 and day 6 (Fig 2C) post-infection compared with WT mice. Viral RNA levels in *c-Flip*$^{+/-}$ mice brains were significantly lower than those in WT mice on day 2 post-infection, rather than on day 6 post-infection (Fig 2D). These results demonstrate that c-FLIP facilitates ZIKV replication in both the periphery and the central nervous system (CNS).

Notably, higher ZIKV mRNA levels were observed in uterus and testis isolated from WT mice compared to *c-Flip*$^{+/-}$ mice on both day 2 and day 6 post-infection (Fig 2E and 2F). To deepen our understanding of c-FLIP's role in ZIKV infection in male mice, we conducted a comparative study of ZIKV replication in male reproductive tissues between WT and *c-Flip*$^{+/-}$ male mice, as described previously [24]. On day 7 post-infection, we observed higher ZIKV replication in the testis, epididymis, and their spermatic liquid in WT male mice (Fig 2G). However, there was no significant difference in sperm activity between ZIKV-infected WT mice and *c-Flip*$^{+/-}$ mice (Fig 2H). These results suggest that c-FLIP promotes ZIKV infection in various reproductive tissues without affecting sperm activity.

## c-FLIP is linked to ZIKV-caused congenital abnormalities of the murine fetus

Considering the decreased ZIKV replication in reproductive organs of *c-Flip*$^{+/-}$ mice, we subsequently explored the impact of c-FLIP knockdown on CZS. At the outset, we chose WT males as mating partners, pairing them with either WT or *c-Flip*$^{+/-}$ females to obtain heterozygous fetuses, given the lethality of homozygous embryos by embryonic days (E) 10.5. Both WT and *c-Flip*$^{+/-}$ pregnant dams received treatment with 2mg MAR1-5A3 on E5.5 to enhance ZIKV

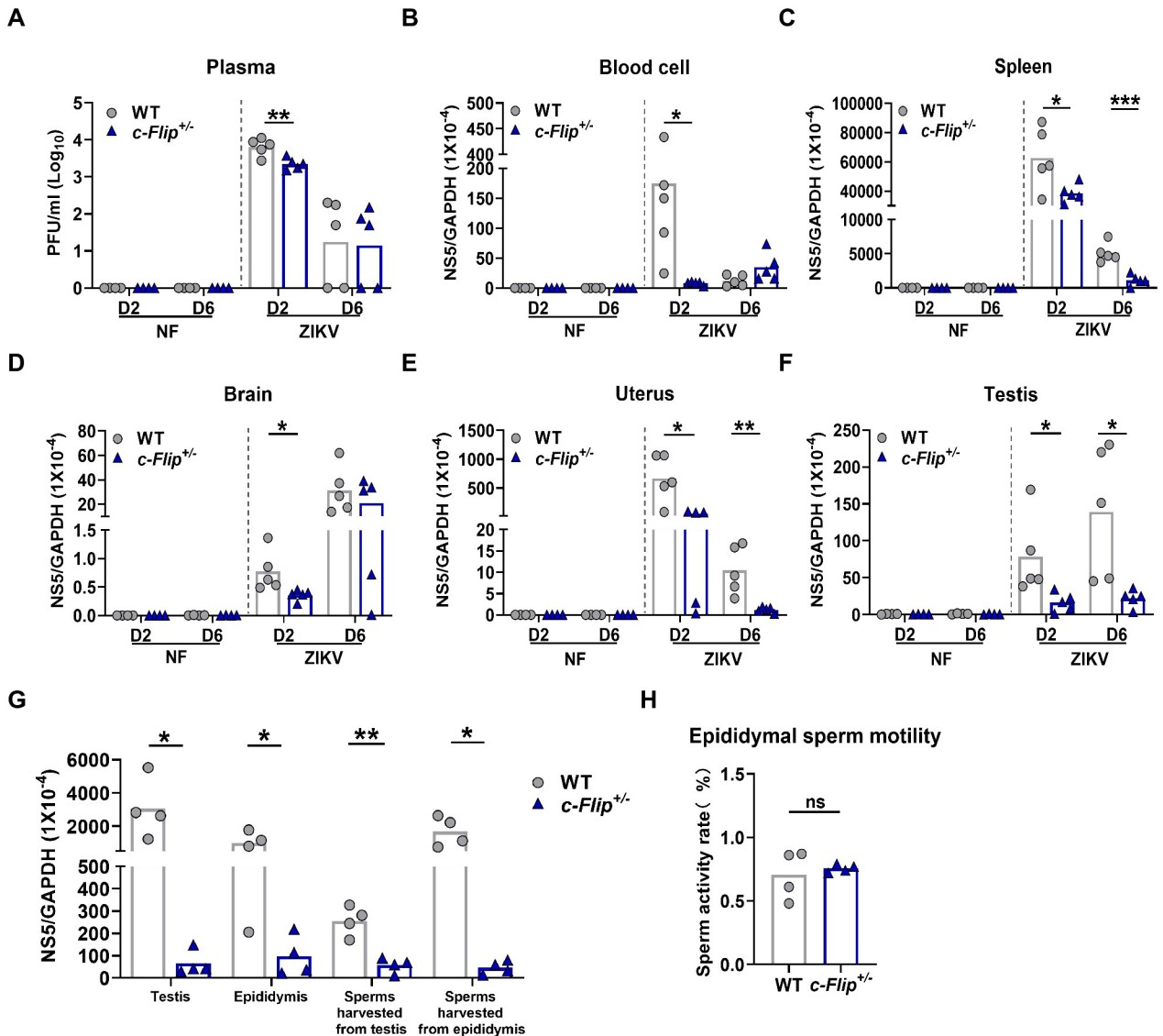

**Fig 2. c-FLIP facilities ZIKV replication in mice.** (A-F) WT and *c-Flip*$^{+/-}$ mice (6-8-weeks-old) were treated with 2 mg MAR1-5A3 on the day prior to infection and then i.p. inoculated with $5 \times 10^5$ PFU of ZIKV. ZIKV replication in plasma (A), blood cell (B), spleen (C), brain (D), uterus (E), and testis (F) was measured on D2 and D6 post-infection by plaque assay or qPCR. (G) WT and *c-Flip*$^{+/-}$ male mice (10-weeks-old) were treated with 2 mg MAR1-5A3 on the day prior to infection and then i.p. inoculated with $1 \times 10^6$ PFU of ZIKV. ZIKV replication in testis, epididymis, and sperm was measured on D7 post-infection by qPCR. (H) The activity of sperm extracted from the epididymis was assessed. The data (n = 5) represent either a single experiment chosen as representative from three independent experiments (A-F) or collective results of 4 mice (G, H). All the data are analyzed by unpaired Student's *t* test. Data are presented as means ± SD. \**P* <0.05, \*\**P* < 0.01, \*\*\**P* < 0.001 compared to control group. ns represents no statistical difference.

permissiveness, followed by challenge with $1 \times 10^7$ PFU ZIKV one day later (Fig 3A). On E13.5, we observed that the fetuses from ZIKV-infected WT group in the uterus underwent fetal demise and subsequent absorption, leaving behind only placental remnants (Fig 3B, Red arrows). Isolated cases of microcephaly were observed in the ZIKV-infected WT group as well as pallor was evident (Fig 3B, Green arrows). However, fetuses from ZIKV-infected *c-Flip*$^{+/-}$ group developed normally with less miscarriage or absorption, and did not exhibit pallor (Fig 3B and 3C). We assessed individual fetuses isolated from the uterus by measuring their weight,

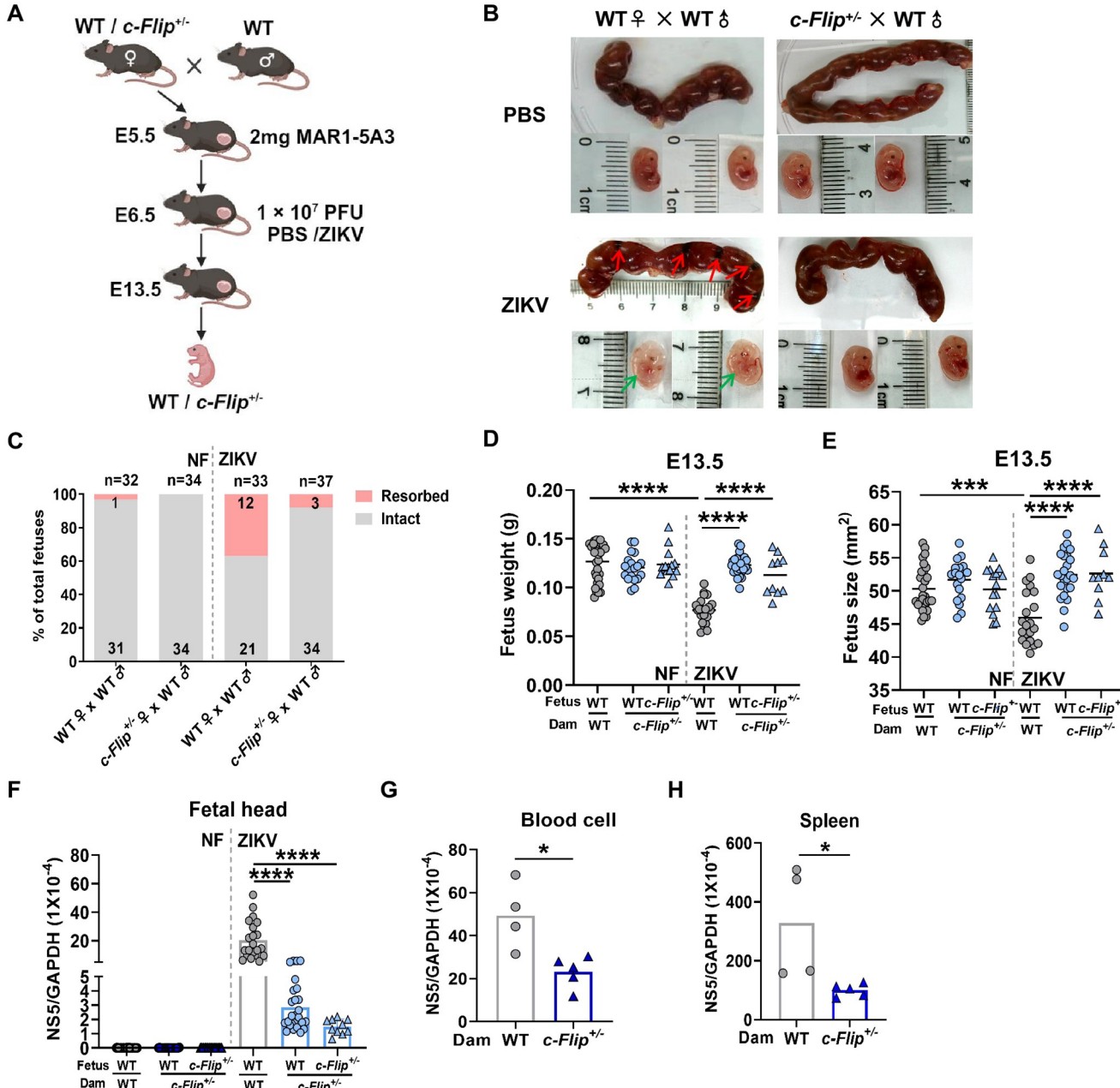

**Fig 3. c-FLIP contributes to ZIKV-induced IUGR.** (A) Schematic representation of the experiment set up. WT or *c-Flip*[+/-] dams mated with WT sires (8-10-weeks-old) were treated with 2 mg MAR1-5A3 on the day prior to infection and then i.p. inoculated on E6.5 with PBS or $1 \times 10^7$ PFU of ZIKV. Placenta and fetal head were harvested on E13.5. (B) Representative images of E13.5 uteri (upper panel) and fetuses (lower panel). Partial demise and growth restriction was shown in ZIKV-infected WT pregnant dams. Red arrows indicate the placenta residues. Green arrows show the growth restriction of fetuses. (C) Resorption rates were analyzed on E13.5. (D-E) The weight (D) and size (E, CRL × OF diameter) of fetuses on E13.5. (F) ZIKV replication in WT and *c-Flip*[+/-] fetal heads of fetuses was measured by qPCR. (G-H) ZIKV replication in the blood cell (G) and spleen (H) of the WT or *c-Flip*[+/-] dams was measured by qPCR. Data are collected from 4–5 pregnant dams per group and analyzed by Turkey test of ANOVA analysis. Data are presented as means ± SD. *$P < 0.05$, ***$P < 0.001$, ****$P < 0.0001$ compared to control group. Fig 3A was created with BioRender.com.

crown-rump length and occipito-frontal (CRL × OF) diameter of the fetal head. Specifically, fetuses from *c-Flip*$^{+/-}$ dams were genotyped, revealing either WT or *c-Flip*$^{+/-}$ status. As controls, we observed no significant difference in weight and size between the WT and *c-Flip*$^{+/-}$ fetuses for the non-infected groups. However, ZIKV infection induced IUGR, as evidenced by weight loss and size reduction in fetuses from WT dams, while such phenomena were not observed in fetuses from *c-Flip*$^{+/-}$ dams (Fig 3D and 3E). Correspondingly, the viral loads in the fetal heads from the ZIKV-infected *c-Flip*$^{+/-}$ group were significantly lower compared to that in WT fetuses from WT dams. ZIKV replication in the WT fetal head was increased compared to the *c-Flip*$^{+/-}$ fetal heads. Intriguingly, a trend of higher ZIKV replication levels in WT fetal heads compared to *c-Flip*$^{+/-}$ fetal heads from the same dam was noted, although this difference did not reach statistical significance (Fig 3F). We subsequently assessed the viral loads in the blood cell and spleen of ZIKV-infected WT and *c-Flip*$^{+/-}$ dams. In line with the outcomes observed in adult mice, *c-Flip*$^{+/-}$ dams exhibited a significant reduction in viral load in both blood cell and spleen compared to WT dams (Fig 3G and 3H). These findings suggest that c-FLIP promotes ZIKV infection and contributes to the enhancement of IUGR in fetuses.

## c-FLIP exhibits a paternal influence on the viral load in the fetal head

Give the lower viral load in *c-Flip*$^{+/-}$ dams might be a reason to affect the vertical transmission, we used WT female mice as dams mated with either WT mice or *c-Flip*$^{+/-}$ mice as sires (Fig 4A). As expected, we observed no significant difference in viral load of blood cells and spleen between dams mated with WT sires and those mated with *c-Flip*$^{+/-}$ sires (S3A and S3B Fig). Notably, we still observed diminished fetal demise and IUGR in *c-Flip*$^{+/-}$ fetuses from ZIKV-infected WT dams (Fig 4B–4E). Although determining the genotype of the placenta in heterozygous dams is complicated [25], we assumed the same genotype for the placenta as that of the fetus since the placenta is derived from extraembryonic tissues [26]. Interestingly, we observed a reduced viral load in the *c-Flip*$^{+/-}$ placenta compared to WT placenta, regardless of whether the sires are WT or *c-Flip*$^{+/-}$ (Fig 4F). This finding suggests a potential role of c-FLIP in influencing the viral load in the placenta. In addition, ZIKV replication was significantly attenuated in the head of both WT and *c-Flip*$^{+/-}$ fetuses from WT dams mated with *c-Flip*$^{+/-}$ sires compared to that in WT fetuses from WT dams mated with WT sires. However, we did not observe a significant difference in viral load between WT and *c-Flip*$^{+/-}$ fetuses from the same WT dams mated with *c-Flip*$^{+/-}$ sires (Fig 4G). Collectively, these results suggest that c-FLIP may exhibit a potential paternal effect on ZIKV replication in the fetal head.

## The regulation of ZIKV infection by c-FLIP involves the mediation of caspase-3 in mice

To elucidate the biological processes contributing to the reduced ZIKV viral load in *c-Flip*$^{+/-}$ fetal heads, we conducted RNA-seq analysis on fetus head homogenates. Gene Ontology (GO) enrichment analysis revealed the top 20 significantly enriched biological processes associated with differential expression of genes (DEGs). In WT fetuses from ZIKV-infected WT dams mated with WT sires, our study revealed a significant enrichment of DEGs within a signaling pathway closely associated with c-FLIP, specifically in the context of apoptosis, when compared to the PBS group (Fig 5A). Furthermore, the additional comparison between ZIKV-infected WT fetuses and *c-Flip*$^{+/-}$ fetuses from WT or *c-Flip*$^{+/-}$ dams also revealed a significant enrichment of DEGs associated with apoptosis or negative regulation of apoptosis (Fig 5B and 5C). Given the reported role of c-FLIP as a regulator of apoptosis mediated by caspase-8 and caspase-3 [15], we sought to investigate whether c-FLIP facilitates ZIKV infection through apoptosis. Subsequently, we compared the activation levels of caspase-8 and caspase-3 in the

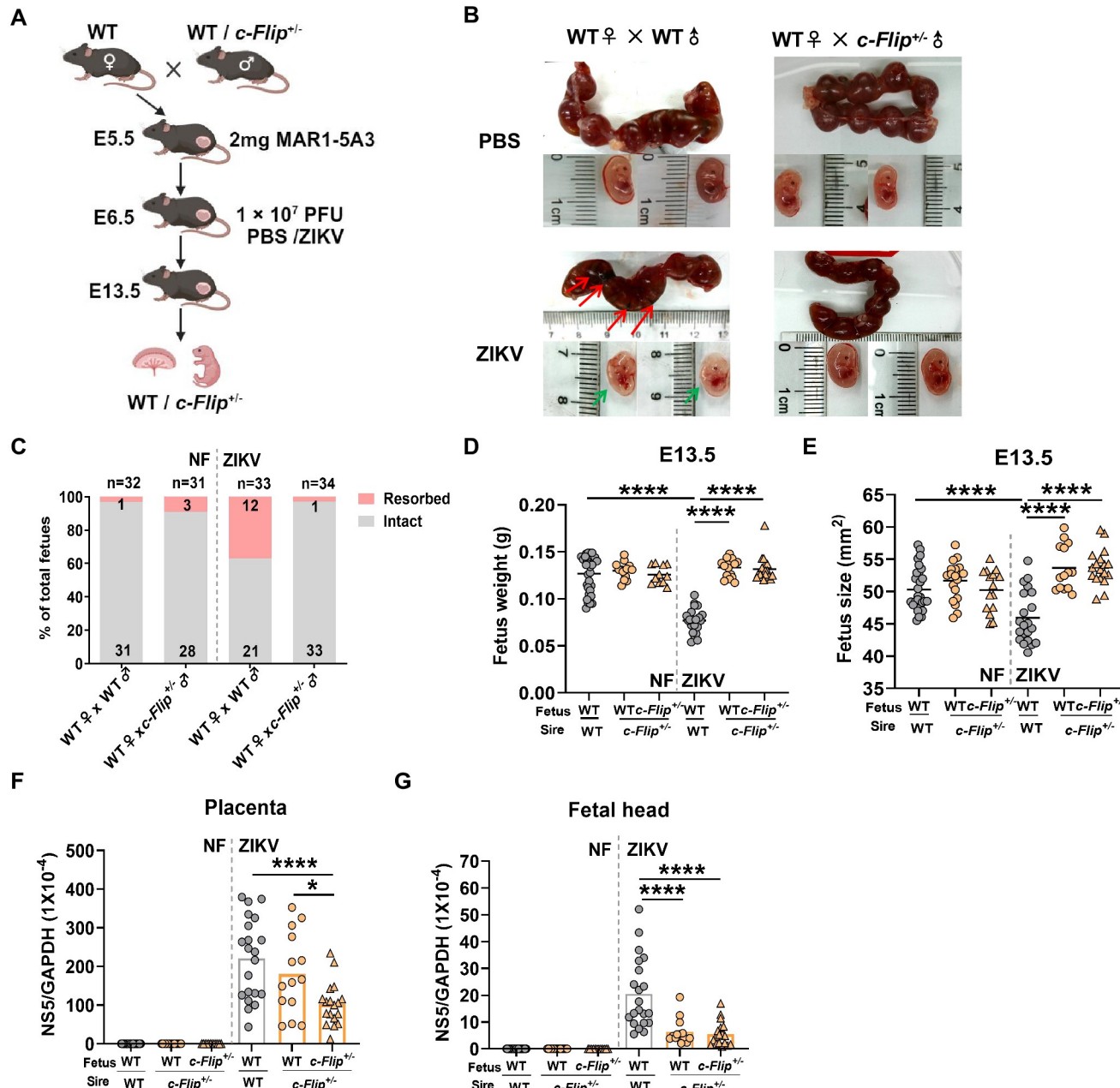

**Fig 4. c-FLIP exhibits a paternal effect on ZIKV replication in the fetal head.** (A) Schematic representation of the experiment set up. WT dams mated with WT or *c-Flip*[+/−] sires (8-10-weeks-old) were treated with 2 mg MAR1-5A3 on the day prior to infection and then i.p. inoculated on E6.5 with PBS or $1 \times 10^7$ PFU of ZIKV. Placenta and fetal head were harvested on E13.5. (B) Representative images of E13.5 uteri (upper panel) and fetuses (lower panel). Partial demise and growth restriction was shown in ZIKV-infected WT pregnant dams. Red arrows indicate placenta residues. Green arrows show growth restriction of fetuses. (C) Resorption rates were measured on E13.5. (D-E) The weight (D) and size (E, CRL × OF diameter) of fetuses on E13.5. (F-G) ZIKV replication in WT and *c-Flip*[+/−] placenta (F) and fetal head (G) was measured by qPCR. Data are collected from 4–5 pregnant dams per group and analyzed by Turkey test of ANOVA analysis. Data are presented as means ± SD. *$P < 0.05$, ***$P < 0.001$, ****$P < 0.0001$ compared to control group. Fig 4A was created with BioRender.com.

fetal head from ZIKV infected WT dams and *c-Flip*[+/−] dams. As shown in Fig 5D and 5E, upon ZIKV infection of the mother, we observed a significant reduction in cleaved caspase-3 levels in both WT and *c-Flip* [+/−] fetuses involving *c-Flip*[+/−] sires or dams, as compared to WT fetuses from mattings between WT parents. However, we did not observe any significant differences

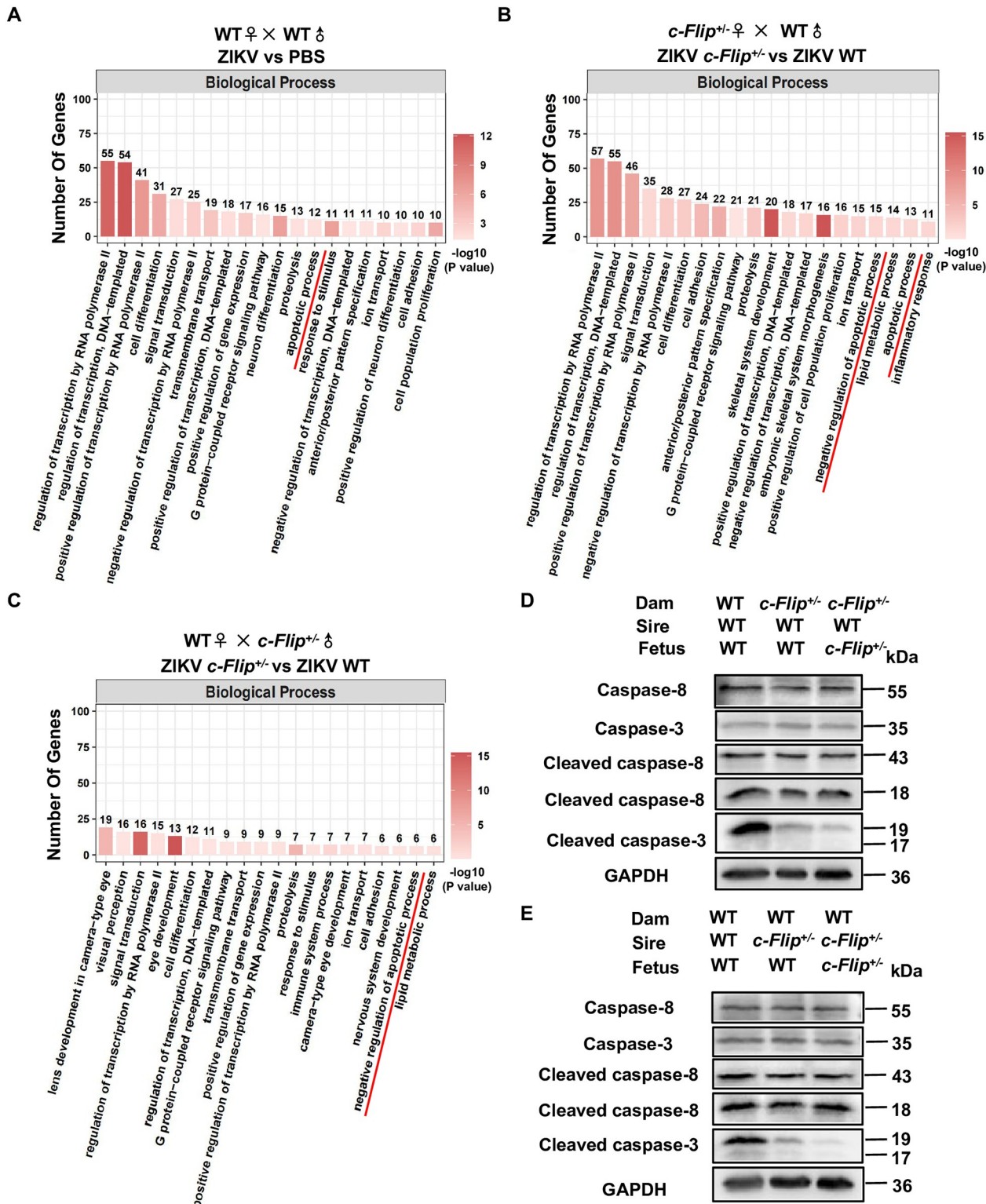

**Fig 5. Caspase-3-mediated apoptosis is involved in c-FLIP regulating ZIKV infection in fetal head.** (A) Functional categorization of DEGs in ZIKV-infected WT fetal head compared with non-infected fetal head of fetus delivered by WT pregnant dam. (B) Functional categorization of DEGs in ZIKV-infected *c-Flip*+/- fetal head compared with WT fetal head of fetuses delivered by *c-Flip*+/- pregnant dam mated with WT sire. (C) Functional categorization of DEGs in ZIKV-infected *c-Flip*+/- fetal head compared with WT fetal head of fetuses delivered by WT pregnant dam mated with *c-Flip*+/- sire. (D-E) Western blot assays of total caspase-8, total caspase-3, cleaved caspase-8 and cleaved caspase-3 expression in ZIKV-

infected WT or *c-Flip*$^{+/-}$ fetal head of fetus delivered by WT or *c-Flip*$^{+/-}$ pregnant dams mated with WT sires (D) or WT pregnant dams mated with WT or *c-Flip*$^{+/-}$ sires (E). DEGs for each biological process in Fig 5A–5C were listed in S1 Table.

in the protein levels of cleaved caspase-8, total caspase-8, and total caspase-3 among the three groups. These results suggest that caspase-3 is involved in c-FLIP regulating ZIKV infection in mice. While our focus was primarily on caspase-8 and caspase-3 due to the known association with c-FLIP and the subsequent activation of effector caspase-3, we acknowledge the significance of other biological processes highlighted by the RNA-seq data. It is important not to overlook these processes in our investigation.

## c-FLIP facilitates ZIKV replication in macrophage through the caspase-8 and caspase-3-mediated apoptosis pathway

To gain a deeper understanding of the mechanism by which c-FLIP regulates ZIKV infection, we assessed the viral load in ZIKV-infected macrophages isolated from both WT and *c-Flip*$^{+/-}$ mice. To enhance permissiveness, the macrophages were pre-treated with MAR1-5A3 followed by ZIKV infection at a MOI of 1 or 3. The viral mRNA levels in *c-Flip*$^{+/-}$ macrophages demonstrated a 40% decrease both on day 1 and day 4 post-infection (MOI = 1), and a 40% decrease on day 1 and 50% decrease on day 4 post-infection (MOI = 3) compared to WT group (Figs 6A and S4A). The viral titers in *c-Flip*$^{+/-}$ macrophages demonstrated a 90% decrease on day 1 and a 70% decrease day 4 post-infection (MOI = 1), and a 50% decrease on day 1 and 60% decrease on day 4 post-infection (MOI = 3) compared to WT group (Figs 6B and S4B). These results indicate c-FLIP promotes ZIKV infection in myeloid cells. To investigate the relationship between c-FLIP and caspase-8/3 in macrophages, firstly, we have confirmed that c-FLIP facilitates caspase-8 and caspase-3 activation upon apoptosis induced by MG132, an apoptosis inducer (Fig 6C). Furthermore, western blot analysis was performed to examine the activation of caspase-8 and caspase-3 in ZIKV-infected WT / *c-Flip*$^{+/-}$ macrophages. As depicted in Fig 6D, there was a reduction in total caspase-8 and caspase-3 protein levels on day 1 (lane 2 vs lane 1) and in full-length caspase-8, full-length caspase-3, and cleaved caspase-3 on day 4 (lane 6 vs lane 5) in *c-Flip*$^{+/-}$ macrophages compared to WT macrophages in the non-infected group. This indicates that c-FLIP may be involved in regulating the expression of caspase-8 and caspase-3 as well as the activation of caspase-3 in myeloid macrophages. Notably, ZIKV infection led to a noticeable increase in cleaved caspase-8 (Fig 6D, lane 7 vs lane 5) on day 4 (S4C and S4D Fig), and in cleaved caspase-3 (Fig 6D, lane 7 vs lane 5) on day 4 (S4E Fig) post-infection in WT macrophages. However, this activation was mostly diminished in the *c-Flip*$^{+/-}$ macrophages (Fig 6D, lane 8 vs lane 7) on day 4 (S4C–S4E Fig) post-infection. Overall, these results indicate that both caspase-8 and caspase-3 are involved in the c-FLIP regulating ZIKV infection in macrophage.

## c-FLIP promotes ZIKV replication in HTR8 cells through the caspase-8 and caspase-3-mediated apoptosis pathway

Next, we expanded our studies to human cells due to our observation of c-FLIP promoting ZIKV replication in HTR8 cells. As shown in Fig 7A and 7B, we identified significant decreases in mRNA levels for both caspase-8 and caspase-3 at the indicated timepoints in c-FLIP knockdown group. As shown in Fig 7C, decreased levels of full-length and cleaved caspase-8/caspase-3 were observed in HTR8 cells transfected with sic-FLIP compared to the control, both in the non-infected group and with ZIKV infection, on day 1 and day 2. Based on the quantification, the ratio of cleaved caspase-8 and caspase-3 to total caspase-8

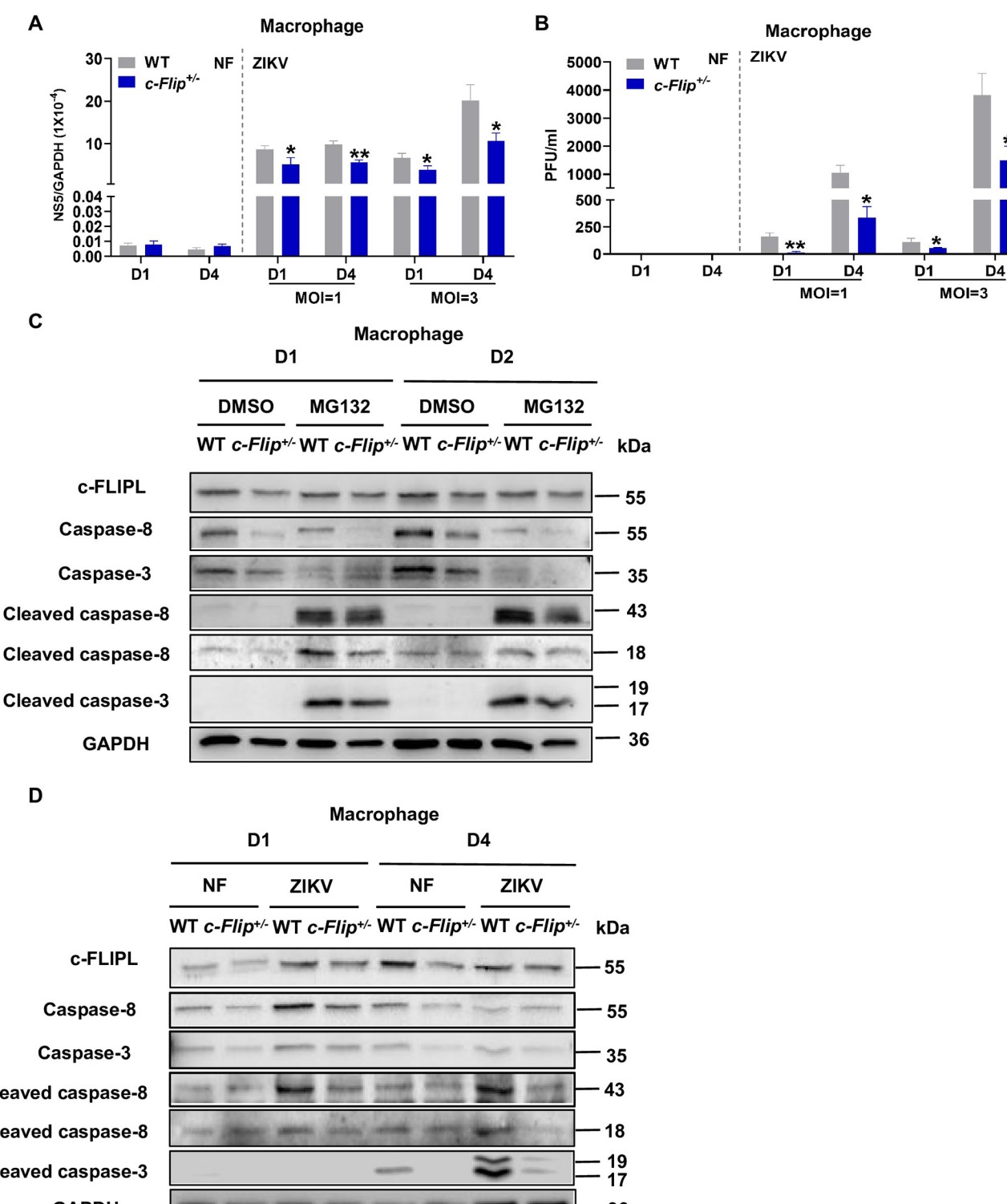

**Fig 6. c-FLIP enhances ZIKV infection in macrophages by activating caspase-8/3.** (A-B) Primary cells isolated from WT or *c-Flip*⁺/⁻ mice were induced to differentiate into macrophages by mouse macrophage colony stimulating factor, incubated with 15μg/mL MAR1-5A3 for 6 h, and then infected with ZIKV at a MOI of 1 or 3. The viral load was measured on D1 and D4 post-infection by qPCR (A) and plaque assay (B). (C) Western blot assays of c-FLIPL, caspase-8, caspase-3, cleaved caspase-8 and cleaved caspase-3 expression in macrophages treated with DMSO or MG132 (10μM) for 1 day and 2 days. (D) Western blot assays of c-FLIPL, caspase-8, caspase-3, cleaved caspase-8 and cleaved caspase-3 expression in macrophages on D1 and D2 post-infection. The data represent a single experiment chosen as representative from three independent experiments and analyzed by unpaired Student's *t* test. Data are presented as means ± SD. *$P < 0.05$, **$P < 0.01$ compared to control group.

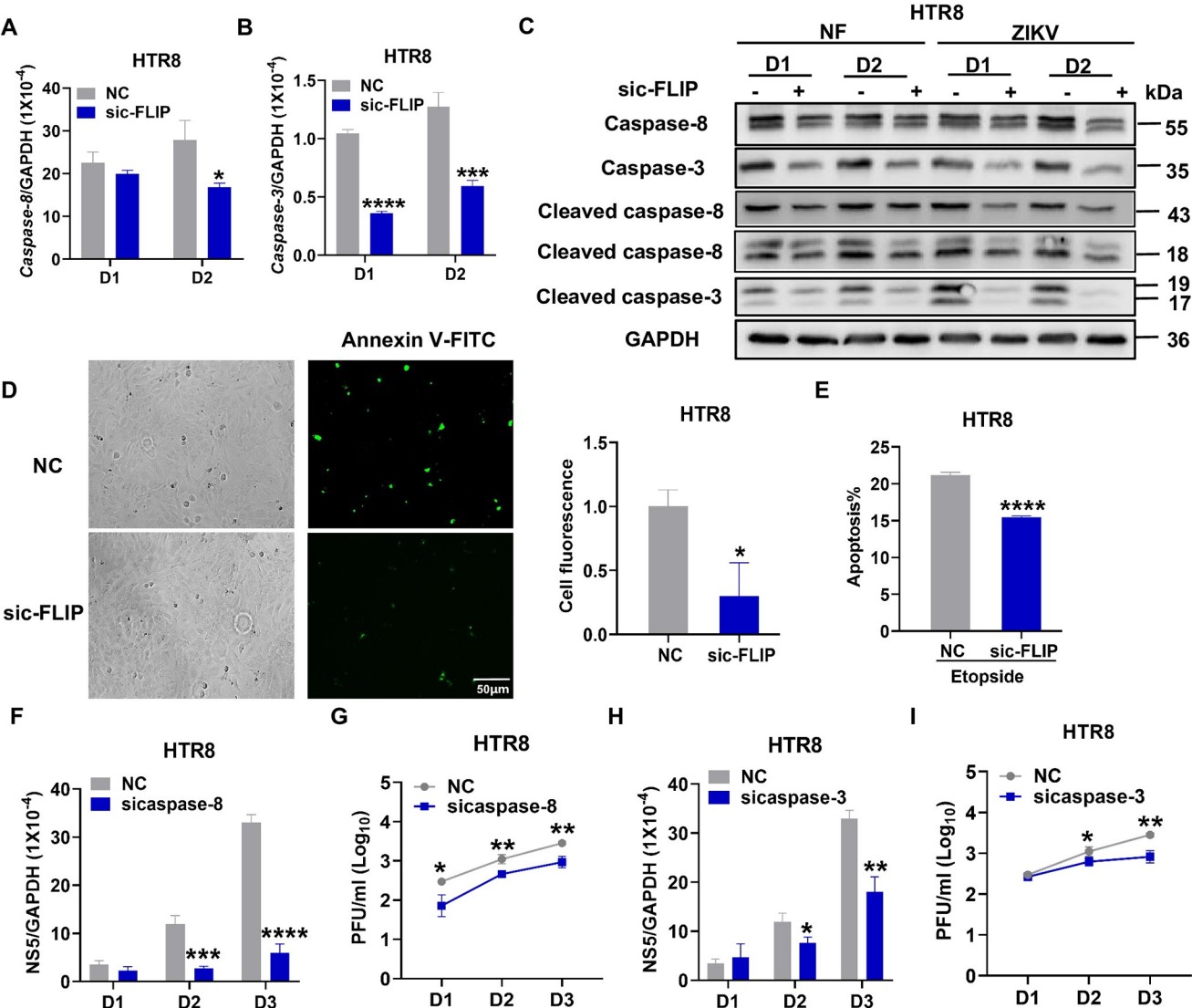

**Fig 7. c-FLIP facilitates ZIKV infection of HTR8 cells by promoting the activation of caspase-8/3.** (A-C) HTR8 cells were infected with ZIKV at a MOI of 1 post sic-FLIP transfection. On D1 and D2 post-infection, caspase-8 and caspase-3 levels were measured by qPCR (A-B) and western blot (C). (D-E) After 1 day of sic-FLIP transfection in HTR8 cells, the cells were either washed with PBS or exposed to etoposide for 4 hours. Subsequently, the cells were incubated with binding buffer containing annexin-V FITC for 15 minutes at 37°C. Finally, cells were stained with annexin V-FITC and observed by confocal microscopy (D) or analyzed by flow cytometry (E). Annexin V-FITC (green) represents cell apoptosis. Scale bar = 50 μm. (F-I) HTR8 cells were infected with ZIKV at an MOI of 1 post caspase-8 and caspase-3 siRNA transfection for 24h. The viral load was measured on D1, D2 and D3 post-infection by qPCR (F, H) and plaque assay (G, I). The data represent a single experiment chosen as representative from three independent experiments and analyzed by unpaired Student's *t* test. Data are presented as means ± SD. *$P < 0.05$, **$P < 0.01$, ***$P < 0.001$, ****$P < 0.0001$ compared to control group.

and caspase-3 is significantly lower in the c-FLIP knockdown group compared to the control group upon ZIKV infection while no significant was observed in non-infected group (S5A–S5C Fig). This indicates reduced caspase-8 and caspase-3 activity with the reduction of c-FLIP expression upon ZIKV infection. Due to the role of c-FLIP in necroptosis and autophagy, we also investigated whether these processes are involved in the regulation of ZIKV infection. As shown in S6A Fig, we observed a decrease in the expression levels of the necroptosis-associated protein pMLKL on day 2 in non-infected HTR8 cells transfected

with sic-FLIP, while ZIKV infection showed no significant effect on the total MLKL and pMLKL protein levels. This suggests that ZIKV does not involve necroptosis in HTR8 cells through the c-FLIP pathway. Notably, there was no significant change in the expression of the autophagy-related protein LC3B. These results indicate that c-FLIP-related necroptosis and autophagy are not involved during ZIKV infection of HTR8 cells. Subsequently, we induced apoptosis in HTR8 cells using etoposide. We then utilized Annexin V labeled with FITC to measure the release of phosphatidylserine to the cell surface induced by apoptosis. Finally, the FITC signal was detected using immunofluorescence assay (IFA) (Fig 7D) and flow cytometry (Fig 7E). The apoptosis is significantly attenuated in HTR8 cells transfected with sic-FLIP (Fig 7D right panel and Fig 7E). We ensured a knockdown efficiency of caspase-8 and caspase-3 at least 50% (S6B–S6E Fig) followed by ZIKV infection. Both ZIKV mRNA levels and titers were significantly reduced in siRNA-transfected HTR8 cells compared to the control (Fig 7F–7I). In addition, we treated HTR8 cells separately with a caspase-8 inhibitor (Z-IETD-FMK) and a caspase-3 inhibitor (Ac-DEVD-CMK TFA). Comparative analysis with the control group revealed a significant reduction in both ZIKV mRNA levels and titers on days 2 and 3 post-infection (S6F–S6I Fig). All these findings collectively indicate that c-FLIP promotes apoptosis in HTR8 cells through the mediation of caspase-8 and caspase-3.

## c-FLIP regulates ZIKV-induced inflammation

As c-FLIP is also a mediator of inflammation [27], we assessed the levels of inflammatory-related cytokines. As depicted in Fig 8A–8J, the inflammatory cytokines TNFα, IL-4, IL-6, IL-13, IL-10, IL-12 (P40), and IL-12 (P70) along with chemokines including CCL2, CCL3, and CCL5, were significantly elevated in the plasma of $c\text{-}Flip^{+/-}$ mice on day 2 post-infection compared that of WT mice. Additionally, a substantial increase in IL-17A, CCL11 and IL-3 were observed in the plasma of $c\text{-}Flip^{+/-}$ mice on day 2 and day 6 post-infection (Fig 8K–8M). These collective findings suggest that c-FLIP plays a negative regulatory role in the majority of the inflammatory response in the plasma of mice.

Furthermore, we observed a significant increase in TNFα, IL-1α, IL-2, IL-3, IL-4, IL-9, IL-10, IL-12 (P70), IL-17A, and CCL5 on day 4 post-infection in murine macrophages (S7A–S7J Fig). CCL-2 exhibited heightened levels on day 1 post-infection, while IL-1β, IL-13 and CCL11 showed increased levels on both day 1 and 4 post-infection, as compared to the non-infected group (S7K–S7N Fig). Conversely, IL-5, IL-6, and IL-12 (P40) experienced a notable decrease on day 4 post-infection, suggesting a distinct regulatory role of c-FLIP in macrophages compared in vivo (S7O–S7Q Fig). Significantly, there was a pronounced reduction in IFN-γ levels in $c\text{-}Flip^{+/-}$ plasma on both days 2 and 6 post-infection, as well as in macrophages on day 4 post-infection (Figs 8N and S7R). This suggests a potential positive regulatory role of c-FLIP in type II interferon in both plasma and macrophages. As previous studies indicate that interferon promotes c-FLIP expression [28], we also treated WT macrophages with interferon. The results show an elevation in the protein expression of c-FLIPL in WT macrophages following treatment with both IFN-α and IFN-γ (S7S Fig), indicating a potential positive regulatory role of c-FLIP for both type I and type II interferons.

Subsequently, we assessed the cytokine levels in ZIKV-infected HTR8 cells. Interestingly, c-FLIP siRNA-transfected HTR8 cells exhibited elevated levels of CCL-7, CXCL1, CXCL12, and TRAIL (S8A–S8D Fig), while showing decreased levels of TNFα, IL-12 (P40), IL-2, IL-16, CCL-5, and MIF compared to the control group (S8E–S8J Fig). These findings collectively indicate the involvement of c-FLIP in ZIKV-induced inflammation and highlight its diverse roles in different cells.

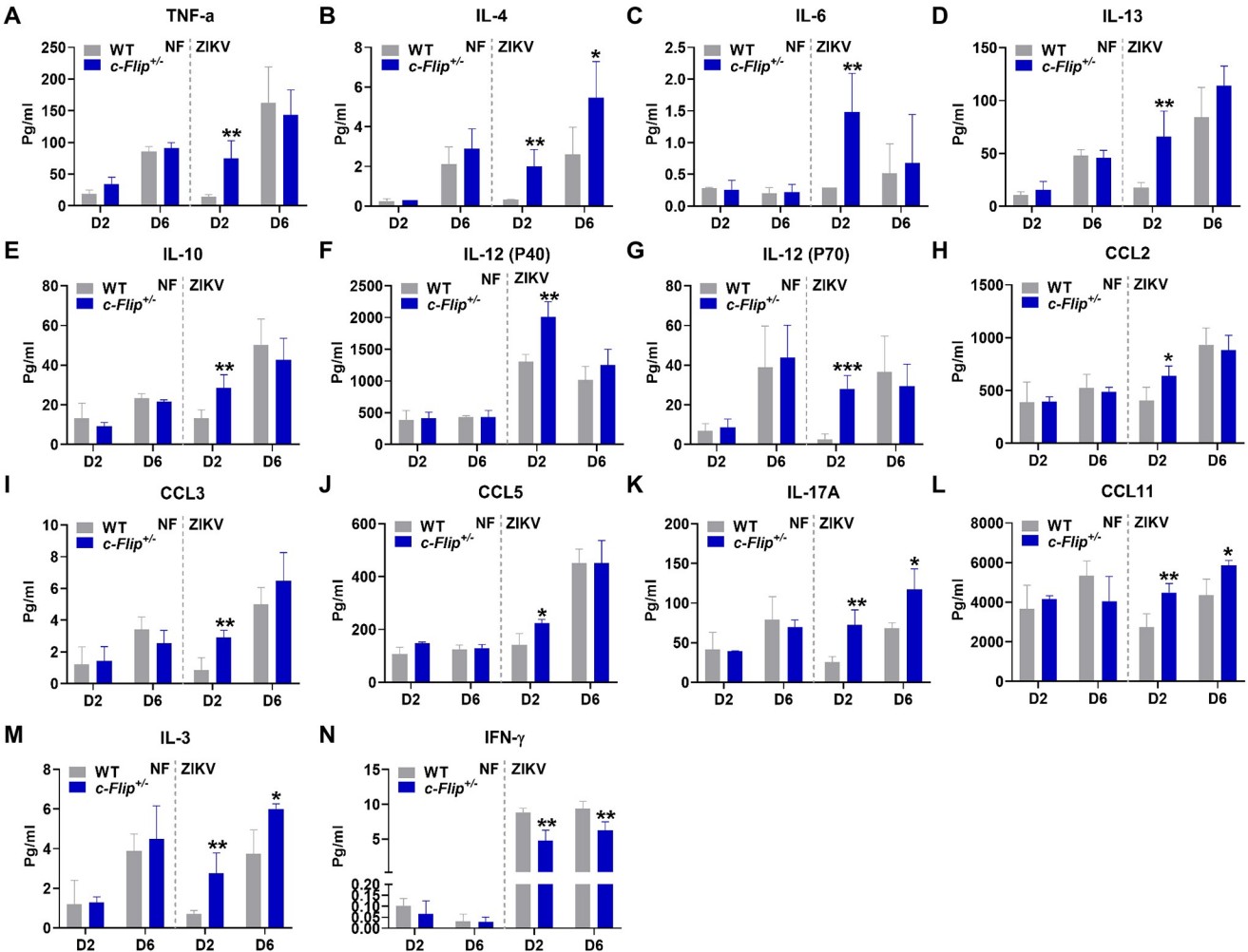

**Fig 8. c-FLIP promotes the production of multiple inflammatory cytokines and chemokines in the plasma.** (A-N) Cytokine levels in plasma of WT or *c-Flip*[+/-] mice (6-8-weeks-old) treated with 2 mg MAR1-5A3 on the day prior to infection and then i.p. inoculated with PBS or $5 \times 10^5$ PFU of ZIKV. Data are collected from 3–4 mice per group and analyzed by unpaired Student's *t* test. Data are presented as means ± SD. *$P < 0.05$, **$P < 0.01$, ***$P < 0.001$ compared to control group.

## Discussion

c-FLIP plays a pivotal role in various biological processes, including apoptosis, programmed necroptosis (necrosis), and autophagy [29]. While the role of c-FLIP during viral infections has been extensively studied in vitro [18–21], in vivo investigations have been constrained by the challenges in obtaining homozygous embryos. Nevertheless, we leveraged heterozygotes to gain insights into c-FLIP's role in promoting ZIKV infection. Our findings in adult mice are compelling, with attenuated ZIKV dissemination from peripheral tissues, including plasma and spleen, to the brain in *c-Flip*[+/-] mice. Most notably, we observed reduced ZIKV replication in the reproductive organs of *c-Flip*[+/-] mice, which correlated with a decrease in fetal CZS.

While c-FLIP was initially regarded as a negative regulator of apoptosis, especially in cancer cells [30,31], subsequent studies have revealed that c-FLIPL can exhibit a pro-apoptotic role under conditions of intense death receptor stimulation, low concentration, or exposure of high concentrations of c-FLIPS or c-FLIPR [31–34]. During the death receptor signal

transduction process, a complex known as the death-inducing signaling complex (DISC) is formed upon the stimulation of a death signal. Both c-FLIP and caspase-8 participate in the formation of DISC. Upon stimulation, DISC is formed, initiating the pro-apoptotic caspase cascade and ultimately leading to cell death [35]. The isoforms of c-FLIP can bind to caspase-8 through its N-terminal death-effector domains (DEDs), and their concentrations subtly regulate caspase-8 activation. c-FLIPL contains two DEDs and a caspase-like domain (CLD). When c-FLIPL acts at low or moderate concentrations, it promotes the formation of heterodimers between c-FLIPL and caspase-8, thereby facilitating the activation of caspase-8 at the DISC. However, at higher concentrations of c-FLIPL, it competitively binds to the DEDs of caspase-8, inhibiting its activation [36]. As the short isoforms, both c-FLIPS and c-FLIPR lacks the CLD domain but retains the DEDs. The DEDs of c-FLIPS are followed by 20 amino acids crucial for its ubiquitination and subsequent targeting for proteasomal degradation. In contrast, c-FLIPR contains two DEDs but lacks the additional carboxy (C)-terminal amino acids present in c-FLIPS [37]. The differential expression of c-FLIPR and c-FLIPS was observed in several cell lines. Furthermore, c-FLIPR and c-FLIPS exhibited comparable biochemical characteristics in their recruitment to the DISC [38]. Although we are greatly interested in studying the roles of different isoforms during ZIKV infection, we were unable to detect all the isoforms and determine their accurate concentrations. We will investigate this further once we obtain an antibody that can detect all the isoforms of c-FLIP. Notably, our primary concern revolves around how c-FLIP enhances ZIKV infection and whether apoptosis is implicated. Multiple studies have established a link between the apoptosis of neural cells in the CNS and microcephaly, both in vitro and in vivo during ZIKV infection. This apoptotic process involves various caspases from the protease family, including caspase-3, -7, -8, and -9, with their specific roles varying depending on the viral strain and cell type [10,39–41]. We chose to focus on caspase-8 and caspase-3 due to the association between c-FLIP and caspase-8, ultimately leading to the activation of effector caspase-3 [42–45]. Notably, during ZIKV infection, we observed reduced activation of caspase-3, rather than caspase-8, in the c-Flip$^{+/-}$ fetal head compared to the WT fetal head. However, in vitro, our findings indicate that c-FLIP regulates ZIKV infection by activating both caspase-8 and caspase-3 in macrophages and HTR8 cells. These results collectively demonstrate that c-FLIP serves as a positive regulator of caspase-8 and caspase-3, promoting ZIKV infection both in vitro and in vivo.

c-FLIP is also implicated in inflammation as a switch between cell death and the host inflammatory response. We observed elevated levels of multiple inflammatory cytokines and chemokines when c-FLIP was partially deficient in mice or murine derived cells. This decreased inflammation, in turn, led to reduced c-FLIP expression, promoting caspase-8-mediated cell death [46]. However, we did observe several pro-inflammatory cytokines reduction including TNFα, IL-12 (P40) with the knock down of c-FLIP in HTR8 cells, indicating a pro-inflammatory role for c-FLIP in placental cells. Besides, type I and type III interferons have been reported to inhibit ZIKV replication [47,48], while type II IFN, IFN-γ, has the opposite effect [49]. In our study, we observed a significant reduction in ZIKV-induced IFN-γ levels in the plasma of c-Flip$^{+/-}$ mice and macrophage compared to WT mice, suggesting a positive regulatory role for c-FLIP in IFN-γ. Prior research has shown that IFN-γ can upregulate caspase-8 expression to facilitate apoptosis [50–52]. Additionally, another study has demonstrated that IFN-γ promotes c-FLIP expression in oligodendrocytes [53], underscoring the intricate interplay between c-FLIP, caspase-8 and IFN-γ.

Notably, the viral load in both fetal heads of WT and c-Flip$^{+/-}$ offspring from WT dams× c-Flip$^{+/-}$ sires were not fully consistent with the viral mRNA level in the placenta. This discrepancy highlights the complex interplay between maternal, placental, and paternal factors in determining fetal outcomes during ZIKV infection. Further investigation into the specific

mechanisms underlying these observations could provide valuable insights into the role of c-FLIP in the vertical transmission of ZIKV and its impact on fetal development.

Overall, our study faces a challenge as obtaining homozygous fetuses is not feasible to elucidate the impact of complete c-FLIP deficiency on ZIKV infection and its vertical transmission. However, we observed that other studies have successfully utilized conditional c-Flip-deficient mice to investigate the role of c-FLIP in the biological process [54,55]. We believe that this approach can offer valuable insights into the regulatory role of c-FLIP in ZIKV infection. Furthermore, given that c-FLIP has been targeted in cancer therapies [56,57], various approved drugs may hold potential for serving as a cure for ZIKV-induced CZS.

## Materials and methods

### Ethics statement

The experiment with infectious ZIKV were conducted under biosafety level 2 (BSL2) facilities at Sun Yat-Sen University and Beijing Laboratory Animal Research Center. The animal experiments were approved by the Animal Care and Use Committee of Sun Yat-sen University.

### Viruses

The Asian lineage ZIKV GZ01 strain (GeneBank accession no: KU820898) was isolated from a Chinese patient who had returned from Venezuela in 2016. The strain, provided by Prof. Jinchun Zhao from Guangzhou Medical University, underwent one or two amplifications in Vero cells. Viral stocks were then stored in aliquots at -80˚C.

### Cells and reagents

The trophoblast cell line (HTR8/SVneo; ATCC, CRL-3271) was purchased from the American Type Culture Collection. These Cells were cultured in RPMI 1640 supplemented with 5% heat-inactivated fetal bovine serum (FBS; CLARK, FB25015) and 100 U/mL Penicillin-Streptomycin (Gibco, 15140–122). The African green monkey (Vero) cell line (ATCC, CCR-81) was provided by Prof. Caijun Sun from Sun Yat-sen University and cultured in Dulbecco's modified Eagle's medium (DMEM; Gibco, C11995500BT) supplemented with 5% heat-inactivated FBS and 100 U/mL Penicillin-Streptomycin. Bone marrow-derived macrophages were isolated using a previously described method [12].

Inhibitor Z-IETD-FMK (MCE, HY-101297) and Ac-DEVD-CMK TFA (MCE, HY-P0034A) were purchased from MedChemExpress. IFN-α (Abclonal, RP01725) protein was provided by Abclonal and IFN-γ (NovoProtein, CM41-50) protein was provided by Abclonal.

### Viral infection

All cell cultures were incubated at 37˚C in a 5% $CO_2$ atmosphere and infected with ZIKV at a multiplicity of infection (MOI) of 1. In the case of macrophages, MAR1-5A3 (15 μg/mL, Leinco Technologies) was added 6 hours prior to ZIKV infection. Supernatants and cells were collected at specified time points post-infection for the analysis of viral loads, cell apoptosis, and cytokine production. Viral titers were determined by performing a plaque assay on Vero cells, following the previously described protocol [58].

### Mice inoculation

The *c-Flip*^+/- mice (on a C57BL/6 background) were generated using CRISPR/Cas9 by Gempharmatech company. The mice were intraperitoneally (i.p.) inoculated with $5 \times 10^5$–$1 \times 10^7$

PFU of ZIKV-GZ01, depending on the experimental requirements. Tissues, embryos, and sperm were harvested as previously described [12,24].

## Genotyping

Mouse tails and fetal bodies were collected for genotype identification using a one-step mouse genotyping kit (Vazyme). Two primer PCR strategies were employed for genotyping: primer set 1 (Forward: 5'-GACATACCAGTTATAGCCCACACAGG-3', Reverse: 5'-GCCTGCTTACTACATGAGACATCAG-3') and primer set 2 (Forward: 5'- AGAG-GATGGCCTTGAAACTTCTG-3', Reverse: 5'- GCATGTGAGTGGAACTGACAAGTC-3'). PCR products were separated on 1.5% agarose gels. For wild-type mice, the targeted alleles exhibited bands at 316 bp. For *c-Flip*$^{+/-}$ mice, the targeted alleles were 648 bp in primer set 1 PCR, and 316 bp and bp in 0 primer set 2 PCR.

## Quantitative real-time PCR (qPCR)

Samples were re-suspended in Trizol (Invitrogen, 15596018) for RNA extraction. cDNA was synthesized by using an iScript cDNA synthesis kit (Bio-Rad, 1708891). The primer sequences for the GAPDH, ZIKV NS5, c-FLIPL, c-FLIPS, c-FLIPR, caspase-3 and caspase-8 genes were listed in S2 Table. The qPCR was performed in the CFX96 real-time PCR system (Bio-Rad, Touch3) with iScript SYBR Green One-Step Kit (Bio-Rad, 1725125). All primers were synthesized by Sangon Biotechnology (Shanghai, China).

## Plaque-forming assay

Each viral stock was used to infect Vero cells in duplicate and viral plaques were visualized 4 days following infection. Briefly, purified virus was diluted at four different 10-fold and then were used to infect monolayer of Vero cells at 37°C for 1 hour, a semi-solid mix of nutriment solution containing 0.8% methyl-cellulose, 0.025M HEPS, 2% fetal bovine serum, 1% Penicillin-Streptomycin, and 0.5% DMSO was added. At 4 days, the nutriment solution was removed. Cells were washed with 1 PBS, fixed with 3.6% formaldehyde for 30 min. Finally, the cells were stained with 1% crystal violet for 5 min. The number of plaques was counted and used to calculate viral titers expressed as PFU/mL.

## Western blot

Proteins were harvested from cells and animal tissues. Proteins were separated and identified as described in detail previously [12]. The following primary antibodies were used: anti-FLIP-human-mAb-NF6 (Adipogen, AG-20B-0056-C100), anti-cleaved caspase-8 antibody (Cell Signaling Technology, 8592), anti-cleaved caspase-3 antibody (Cell Signaling Technology, 9664), anti-caspase-8 antibody (Proteintech, 13423-1-AP), anti-caspase-3 antibody (Proteintech, 19677-1-AP), anti-Zika virus NS5 antibody (Genetex, 133312), anti-MLKL antibody (Affinity Biosciences, DF7412), anti-pMLKL antibody (Abcam, ab196436), anti-LC3B antibody (Cell Signaling Technology, 3863S).

## Cell viability assay

$2\times10^4$ cells were inoculated in each well of 96-well-plate, and a series of concentrations of NC or siRNA were added. After incubating for 24 hours, 10μl CCK8 (Biosharp, CCK8-BS350B) solution was added into each well, followed by 2 hours in the dark at 37°C. The OD at 450 nm was then detected by a microplate reader (Biotek, Synergy HTX, USA)

## Sperm activity assessment

After dissecting male mice, sperm were extracted from the epididymis and testicles, and then placed in M2 medium for incubation at a temperature of 37˚C for 10 minutes to facilitate sperm capacitation. Take 10 μL of suspended semen and place it in a 1.5 mL EP tube. Add 90 μL of sterile $H_2O$ to achieve dilution. Homogenize the mixture and carefully transfer 10 μL to the blood cell counting plate, allowing the semen to flow naturally into the counting chamber for enumeration. To initiate the calculation, first determine the X1 value of deceased sperm. Next, position the counting plate within an enamel dish and expose it to a 50˚C water bath for 10 minutes to terminate the sperm. Determine the total sperm count, denoted as X2. Calculate sperm motility as (X2—X1) divided by X2. Ensure that the sperm count is conducted three times per sample [59].

## Cytokine Bio-Plex

Cell culture supernatants and mouse plasma were collected form for analysis of cytokine production using Bio-Plex Pro-Human Cytokine 48-plex Assays (Bio-Rad, 12007283) and Bio-Plex Pro-Mouse Cytokine 23-plex Assays (Bio-Rad, M60009RDPD), respectively. Add 20μL sample (including diluted mouse plasma, serum and HTR8 cells supernatant) and 35μL beads to each well, then incubate at 37˚C for 1 hour. Following the incubation, wash the beads by adding 400μL wash buffer. Subsequently, add 20μL of detection antibody and incubate at 37˚C for 35 minutes, followed by another wash of the magnetic beads. Next, add 50μL Bio-Plex SA-PE (Streptavidin-Phycoerythrin) and incubate for 15 minutes, and wash the beads once more. Utilize the Bio-Plex 200 System to read the data, and Bio-Plex Software presents data as median fluorescence intensity (lMFl) as well as concentration (pg/ml). The concentration of the cytokines and chemokines bound to each bead correlates with the reported signal MFI. Finally, employ the Bio-Plex Data software for data analysis.

## siRNA transfection

Endogenous c-FLIP, caspase-3 and caspase-8 expression were knocked down using siRNA (sic-FLIP: 5′-GGUUGAGUUGGAGAAACUAdTdT-3′ [60], sicaspase-8: 5′-AACCUCGG-GAUACUGUCUGA-3′ [61], sicaspase-3: 5′-UGGAUUAUCCUGAGAUGGG-3′ [62]. Monolayers of HTR8 cells in 24/6-well cell culture plates were transfected with 16.5/66.5 pmoL siRNA and negative control (NC) per well using jetPRIME transfection reagent (Polyplus, 101000046).

## Apoptosis assay

The apoptosis ratio was assessed using the annexin V-fluorescein isothiocyanate (FITC) assay (Beyotime, C1062). At 24 hours post-transfection, cells were either washed with 1× PBS or exposed to etoposide (MCE, HY-13629) for 4 hours, followed by incubation with 500 μL binding buffer containing 5 μL annexin-V for 15 minutes at room temperature in the dark. Annexin V-FITC detection were performed using the LSR IV cytometer Cytoflex S (Beckman Coulter) and the Keyence BZ-X800LE scanning confocal microscope, respectively.

## RNA-sequence

Total RNA from fetal head tissue was isolated and purified using Trizol following the manufacturer's protocol. Poly(A) RNA was purified from 1 μg total RNA using Dynabeads Oligo(dT) 25-61005 (Thermo Fisher, CA, USA) with two rounds of purification. The RNA was then reverse-transcribed into cDNA using Reverse Transcriptase (Invitrogen, USA). Subsequently,

cDNA sequencing was performed on an Illumina 6000 platform by LC-Bio Technologies (Hangzhou, China). StringTie was employed for mRNA expression level quantification, calculated as FPKM (FPKM = [total_exon_fragments/mapped_reads (millions) × exon_length (kB)]). Differentially expressed mRNAs were selected with a fold change > 2 or fold change < 0.5 and analyzed using a parametric F-test comparing nested linear models ($P$ value < 0.05) with the R package edgeR.

### Statistic

All data were analyzed using GraphPad Prism software (version 8.0.2). The quantification of western blot was conducted using Image J. The results are presented as means ± standard deviation (SD). Statistical significance is determined using unpaired Student's $t$ test or Turkey test of ANOVA analysis, and $P$-values are indicated as follows: * For $P$ values < 0.05, ** for $P$ values < 0.01, *** for $P$ values < 0.001, and **** for $P$ values < 0.0001.

The numerical data used in all figures are included in S1 Data.

### Supporting information

**S1 Fig. The efficiency of siRNA knockdown for c-FLIP.** (A) HTR8 cells were infected with ZIKV at a MOI of 1. c-FLIPR levels were measured on D1, D2, D4 and D6 post-infection by qPCR. (B) c-FLIPL, c-FLIPS and c-FLIPR were measured by qPCR post sic-FLIP transfection on day 1. (C) Western blot of c-FLIPL and c-FLIPS expression in HTR8 cells post sic-FLIP transfection on day 1 and day 2 were assessed. (D) Quantification of c-FLIPL and c-FLIPS protein levels relatives to GAPDH. (E) HTR8 cells were transfected with various concentrations of negative control (NC) or sic-FLIP for 1 day, followed by the detection of cell viability using the CCK8 assay. (F) The fold change in Fig 1F. The data are one representative of three independent experiments and analyzed by unpaired Student's $t$ test. Data are presented as means ± SD. *$P$ <0.05, **$P$ <0.01, ****$P$ < 0.0001 compared to control group.
(TIF)

**S2 Fig. Establishment of the *c-Flip*$^{+/-}$ mouse model.** (A) The *c-Flip*$^{+/-}$ mice were generated by using CRISPR/Cas9 to knockout exon 3–5 of c-FLIP gene. (B) PCR analysis of genomic DNA confirmed the genotypes of WT and *c-Flip*$^{+/-}$ mice. The first and second lanes depict gene fragments of the c-FLIP chromosome in WT mice, with sizes of 316bp and 6621bp, respectively. The third and fourth lanes show gene fragments of c-FLIP chromosome in *c-Flip*$^{+/-}$ mice, with sizes of 316bp and 648bp, respectively. (C-F) Western blot assays of c-FLIPL expression in spleen (C), brain (D), uterus (E), and testis (F) of the WT and *c-Flip*$^{+/-}$ mice. S2A Fig was created with BioRender.com.
(TIF)

**S3 Fig. ZIKV mRNA levels in the WT dams.** (A-B) ZIKV replication in the blood cell (A) and spleen (B) collected from 4–5 pregnant dams per group was measured by qPCR. Data are analyzed by unpaired Student's $t$ test and presented as means ± SD. The data represent the collective results of three independent experiments. All the data are analyzed by unpaired Student's $t$ test. Data are presented as means ± SD. ns indicates a non-significant difference.
(TIF)

**S4 Fig. The ratio of cleaved caspase-8/3 to total caspase-8/3 in macrophages.** (A-B) The fold change analysis in Fig 6A and 6B. (C-E) Quantification of the western blot in Fig 6D was conducted. The ratio of cleaved caspase-8/3 to total caspase-8/3 was measured by comparing the protein bands between cleaved caspase-8 and total caspase-3 corresponding to each time point. The data represent either a single experiment chosen as representative from three

independent experiments (A-B) or the collective results of three independent experiments (C-E). All the data are analyzed by unpaired Student's *t* test. Data are presented as means ± SD. \*$P$ <0.05, \*\*$P$ < 0.01, \*\*\*$P$ < 0.001, \*\*\*\*$P$ < 0.0001 compared to control group.
(TIF)

**S5 Fig. The ratio of cleaved caspase-8/3 to total caspase-8/3 in HTR8 cells.** (A-C) Quantification of the western blot in Fig 7C was conducted. The ratio of cleaved caspase-8/3 to total caspase-8/3 was measured by comparing the protein bands between cleaved caspase-8 and total caspase-3 corresponding to each time point. The data represent the collective results of three independent experiments. All the data are analyzed by unpaired Student's *t* test. Data are presented as means ± SD. \*$P$ <0.05, \*\*$P$ < 0.01, \*\*\*$P$ < 0.001 compared to control group.
(TIF)

**S6 Fig. Caspase-8 and caspase-3 involves in ZIKV replication.** (A) HTR8 cells were infected with ZIKV at a MOI of 1 post sic-FLIP transfection. On D1 and D2 post-infection, MLKL, pMLKL and LC3B levels were measured by western blot. (B-C) Caspase-8 (B) and caspase-3 (C) were measured by qPCR post sicaspase-8 or sicaspase-3 transfection on day 1. (D-E) Western blot assays of caspase-8 (D) or caspase-3 (E) expression in HTR8 cells post sicaspase-8 or sicaspase-3 transfection for 1 day and 2 days. (F-I) HTR8 cells were infected with ZIKV at a MOI of 1 post sicaspase-8 (F-G) or sicaspase-3 (H-I) transfection for 24 hours. The viral load was measured on D1, D2 and D3 post-infection by qPCR (F, H) and plaque assay (H, I). The data represent either a single experiment chosen as representative from three independent experiments (B-C, F-I) or the collective results of three independent experiments (D-E). All the data are analyzed by unpaired Student's *t* test. Data are presented as means ± SD. \*$P$ <0.05, \*\*$P$ < 0.01, \*\*\*$P$ < 0.001, \*\*\*\*$P$ < 0.0001 compared to control group.
(TIF)

**S7 Fig. c-FLIP influences inflammatory cytokine production induced by ZIKV in macrophages.** (A-R) Cytokine levels in the supernatant of ZIKV infected WT or *c-Flip*$^{+/-}$ macrophages (MOI = 1). Data are analyzed by unpaired Student's *t* test and are presented as means ± SD. \*$P$ <0.05, \*\*$P$ < 0.01, \*\*\*$P$ < 0.001, \*\*\*\*$P$ < 0.0001 compared to control group. (S) WT macrophages were treated with PBS, IFN-α (20ng/ml) and IFN-γ (20ng/ml) respectively, and the expression level of c-FLIPL was measured by western blot.
(TIF)

**S8 Fig. c-FLIP affects ZIKV induced inflammatory cytokine production in HTR8 cells.** (A-J) HTR8 cells were infected with ZIKV at an MOI of 1 post sic-FLIP transfection for 24h. Cytokine levels in the supernatant of HTR8 cells were measured on D3 post-infection. Data are analyzed by unpaired Student's *t* test and presented as means ± SD. \*$P$ <0.05, \*\*$P$ < 0.01, \*\*\*$P$ < 0.001, \*\*\*\*$P$ < 0.0001 compared to control group.
(TIF)

**S1 Table. DEGs for each biological process in Fig 5A–5C.**
(XLS)

**S2 Table. Primers for qPCR used in this study.**
(DOCX)

**S1 Data. Contains raw data that was used to construct Figs 1, 2, 3, 4, 6, 7, 8, S1, S3, S4, S5, S6, S7 and S8.**
(XLSX)

## Acknowledgments

We are grateful to Prof. Jinchun Zhao (Guangzhou Medical University) for providing the ZIKV GZ01 strain and Prof. Mang Shi (Sun Yat-sen University, China) for their constructive advice on research design and refinement. We thank the Experimental Teaching Center, School of Public Health, Shenzhen (Sun Yat-sen University, China) and Biosafety level 2 (BSL-2) laboratory (Sun Yat-sen University, China) for use of their facilities and services.

## Author Contributions

**Conceptualization:** Nina Li, Huanle Luo.

**Formal analysis:** Shengze Zhang, Nina Li.

**Funding acquisition:** Xuan Zou, Mang Shi, Caijun Sun, Yuelong Shu, Huanle Luo.

**Investigation:** Jiani Wu, Shike Zeng, Lin Zhu, Shaohui Bai, Haolu Zha, Weijian Tian, Nan Wu, Chuming Luo.

**Methodology:** Shengze Zhang, Nina Li, Shu Wu, Ting Xie, Qiqi Chen.

**Project administration:** Huanle Luo.

**Resources:** Shisong Fang, Yuelong Shu, Huanle Luo.

**Supervision:** Huanle Luo.

**Visualization:** Shengze Zhang, Nina Li.

**Writing – original draft:** Shengze Zhang, Nina Li, Huanle Luo.

**Writing – review & editing:** Shengze Zhang, Huanle Luo.

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
