## [Decision Letter · Decision Letter 0]

26 Feb 2024

Dear Mis Luo,

Thank you very much for submitting your manuscript "c-FLIP facilitates ZIKV infection by mediating caspase-8/3-dependent apoptosis" for consideration at PLOS Pathogens. As with all papers reviewed by the journal, your manuscript was reviewed by members of the editorial board and by several independent reviewers. In light of the reviews (below this email), we would like to invite the resubmission of a significantly-revised version that takes into account the reviewers' comments.

While the authors found that there is merit in this study, all reviewers felt that additional experimentation and data analysis are required to more robustly support the conclusions of the manuscript.

In particular, the reviewers asked for new experiments to clarify the following points:

(1) Further experiments and controls are required to more robustly confirm the interaction between c-FLIP and apoptotic and interferon pathways.

(2) Additional western blot controls, and quantification of all western blot data, with multiple replicates and statistical analysis, are required to strengthen the findings.

(3) More detailed analysis of RNA-Seq data is required to put the follow-up experiments into context, and consideration of potential other mechanisms and pathways should be provided in the Discussion.

(4) The appropriateness of statistical tests was questioned, and details of statistical analyses need to be provided. Careful consideration should be given to all statistical tests applied throughout the work.

Please also ensure that the figure resolution meets the journal guidelines, and that figures and wording are checked for maximum clarity throughout (bearing in mind the reviewers' suggestions).

We cannot make any decision about publication until we have seen the revised manuscript and your response to the reviewers' comments. Your revised manuscript is also likely to be sent to reviewers for further evaluation.

Sincerely,

Kevin Maringer, PhD

Guest Editor

PLOS Pathogens

Michael Letko

Section Editor

PLOS Pathogens

Michael Malim

Editor-in-Chief

PLOS Pathogens

orcid.org/0000-0002-7699-2064

While the authors found that there is merit in this study, all reviewers felt that additional experimentation and data analysis are required to more robustly support the conclusions of the manuscript.

In particular, the reviewers asked for new experiments to clarify the following points:

(1) Further experiments and controls are required to more robustly confirm the interaction between c-FLIP and apoptotic and interferon pathways.

(2) Additional western blot controls, and quantification of all western blot data, with multiple replicates and statistical analysis, are required to strengthen the findings.

(3) More detailed analysis of RNA-Seq data is required to put the follow-up experiments into context, and consideration of potential other mechanisms and pathways should be provided in the Discussion.

(4) The appropriateness of statistical tests was questioned, and details of statistical analyses need to be provided. Careful consideration should be given to all statistical tests applied throughout the work.

Please also ensure that the figure resolution meets the journal guidelines, and that figures and wording are checked for maximum clarity throughout (bearing in mind the reviewers' suggestions).

Reviewer's Responses to Questions

**Part I - Summary**

Reviewer #1: The authors report a role for c-FLIP, an apoptosis regulator, in ZIKV induced developmental defects. The work is of general importance, potentially linking previous findings from the group on the role of the E3 ligase Peli1, of which c-FLIP is a target. Overall the studies are well presented and the paper is well written. The work is a logical extension of the previous reported findings. The authors use cell culture models for some of the mechanistic work and because a homozygous KO animal was deemed not obtainable due to previous reports a heterozygous mouse model was used. Overall the work could be of interest to Plos Pathogens readers although some of the claims should be tempered unless further experiments are provided.

Reviewer #2: In this manuscript, Zhang et al. describe how the cellular FLICE-like inhibitory protein (c-FLIP) facilitates Zika virus (ZIKV) infection, playing a potential role in the extrinsic pathway of apoptosis via caspase-8/3. This work has novelty demonstrating that c-FLIP promotes ZIKV infection in placental cells, myeloid-derived macrophages, and other multiple mice tissues, including blood cells, the spleen, the uterus, the testis, and the brain. However, the critical role of c-FLIP as a positive regulator in caspase-8/3-mediated apoptosis during ZIKV infection needs further investigation. In addition, other clarifications are required.

Reviewer #3: This is an interesting study by Zhang and colleagues evaluating the impact of c-FLIP on Zika virus replication. C-FLIP has been previously shown to regulate apoptosis and inflammation and thus the authors pursue the hypothesis that this factor might underlie some of the complications seen during congenital Zika syndrome. Using immortalized trophoblasts, primary macrophages, and murine in vivo models, the authors show that c-FLIP knockdown reduces Zika virus replication, protects embryos from Zika congenial syndrome, reduces the levels of cleaved caspase-3 and -8, and alters the expression of several proinflammatory cytokines. The manuscript is well written, the work is novel and provides important descriptive work into potential determinants of Zika congenital syndrome – which is of great interest - using a novel mouse model. Please see below some comments and suggestions to strengthen the conclusions resulting from this work.

**Part II – Major Issues: Key Experiments Required for Acceptance**

Reviewer #1: 1) A major conclusion of the paper is that c-FLIP enhances ZIKV replication through caspase3/8 dependent mechanisms. The authors should perform an experiment with a caspase 3 and/or 8 specific inhibitor to support their conclusion as off target effects of caspase knock down cannot be excluded with teh present data.

2) Western blots for full length caspase 3 and 8 should be included in all appropriate figures.

3 Western blots for c-FLIP levels in different tissues should be included.

Reviewer #2: Major concerns:

1. Fig. 1A and Fig. 1B: normalisation for the y-axis is required. Assessing fold-change differences between infected and non-infected cells and between c-FLIPL and c-FLIPS is very difficult.

2. Fig. 1C: it is indicated in the text that “these results indicate an augmentation of c-FLIPL and c-FLIPS expression in ZIKV-infected HTR8 cells”. However, this is very difficult to assess with Fig. 1C, and some quantification, at least in triplicate, is required.

3. In S1A, why are the levels for c-FLIPL and c-FLIPS so different from Fig. 1A?

4. In S1B, why are protein levels for c-FLIPL in non-treated cells lower at 48h compared to 24h? In Fig. 1C, these levels are increasing with time. In addition, protein quantification is required.

5. It is stated in the text that the homozygous c-FLIP-/- genotype is lethal. How is the cell viability in silenced cells? Can this be assessed to discard the reduction in virus replication due to lower cell viability?

6. Assessing differences in c-FLIP expression in the different mouse tissues would be very interesting. It could explain why partial knock-down of c-FLIP in the spleen is more detrimental to virus replication than in the brain. Also, is this related to the different isoforms? How are the different isoforms expressed in the different tissues?

7. The information about partial demise and growth restriction in ZIKV-infected WT pregnant dams is very scarce in the text. More clarification is required to understand the presented results.

8. The list of enriched genes by RNA-Seq presented in Fig. 5A is required for each group to assess which genes have been included in each category.

9. In Figs. 5D, 5E and 6C, several panels for other viral and cellular proteins are required to strengthen the results. First, a comparison with non-infected animals is needed. This also needs to include a blot for a viral protein. In addition, it is only possible to assess how much caspases 8 and 3 have been cleaved with the blots for the uncleaved caspases. This will also require quantification in multiple replicates.

10. In Fig. 6, it is stated that “c-FLIP positively regulates caspase-8 and caspase-3 mediated apoptosis”; however, this is overstated without repeating these experiments with a positive control for apoptosis induction independently of virus infection.

11. Fig. 6A and 6B show minimal differences between WT and c-FLIP-/- macrophages, probably due to poor infection. Is it possible to repeat these infections in more permissive cell lines where cFLIP can be silenced, or can this be repeated with a higher MOI?

12. In Fig. 7C, quantifying the uncleaved/cleaved caspase ratio is necessary to discern whether a lower amount of active caspase 8/3 is due to less apoptosis or lower mRNA levels.

13. In Fig. 7D and E, it needs to be better explained how apoptosis has been measured by flow cytometry and immunofluorescence. What is etoposide and annexin-V measuring?

14. In S4A and S4B, according to the Western blot, the knockdown efficiency of the different caspases seems to be less than 50%. Can this be quantified and provide mRNA levels of these transcripts?

15. In Fig. 7G and I, the viral titres in HTR8 non-silenced cells are very low compared to those in Fig. 1F. How can this discrepancy be assessed?

16. In Fig. 8, indicate in the text how the cytokines and chemokines levels have been assessed.

17. Fig. 8N and S5R: the potential positive regulatory role of c-FLIP in type II IFN can be further investigated by adding IFN lambda to cells.

18. The mechanism of how c-FLIP influences the extrinsic apoptosis pathway upon infection must be further discussed regarding the extensive published literature. For instance, how c-FLIP interacts with pro-caspase 8. Also, it will be interesting to know how c-FLIP is upregulated during infection. Is this only at the transcriptional level or also at the post-translational level?

19. c-FLIP has an influence not only on apoptosis but also on necroptosis and autophagy. Are there differences in the intrinsic apoptosis pathway, necroptosis or autophagy in c-FLIP silenced cells upon ZIKV infection?

20. The interplay of the different c-FLIP isoforms during ZIKV infection and apoptosis activation should be further discussed.

21. Have the authors planned to repeat the most relevant experiments with an isolate of the African lineage? This will strengthen the observed results or clarify differences in virulence between lineages.

Reviewer #3: 1. Some critical controls are missing. For instance, the authors conclude that c-FLIP promotes zika virus infection by modulating caspase-3/-8 apoptosis. While the authors show decreased levels of cleaved caspase-3 and -8, the authors should also include the levels of pro-caspase-3 and -8 levels (Figures 5D, E, 6C, and 7C) to support this claim.

2. In Figure 5 A-C, the authors conduct gene ontology analyses of genes that are differentially expressed as a result of Zika infection both in WT and in c-Flip+/- cells. In my opinion, it will be more informative to compare genes that are differently expressed as a result of c-FLIP expression (comparing WT versus c-Flip+/- cells) to understand the impact of c-Flip knockdown in the transcriptome and in response to Zika infection.

3. In Figure 1B, the authors compare the expression of c-FLIP in non-infected versus Zika-infected cells over 6 days in the human cell line HTR8. It is surprising that the levels of c-FLIP increase over time by 3-fold even in the non-infected conditions. One explanation is that c-FLIP is regulated by autophagy (and activated due to media starvation), which in turn can regulate interferon stimulated genes (ISGs). It will be important to show that c-FLIP is not an ISG, given that the in vivo and ex vivo results were conducted in the presence of an interferon signalling neutralizing antibody.

**Part III – Minor Issues: Editorial and Data Presentation Modifications**

Reviewer #1: Figure 1a. The authors claim a significant increase in c-FLIPL with ZIKV infection, but this is only moderately apparent at one timepoint, and the authors ignore that c-FLIPL increases in expression throughout the timecourse in the non-infected control. The authors should acknowledge and comment on this in their presentation of the results.

Figure S1. The western blots are very low resolution and so difficult to interpret. However it appears there is a major effect of the non targeting control siRNA on teh expression of c-FLIPL between 24 and 48 hours. This should be quantified and reported.

Figure 1. The authors swap between days and hours for treatment in the different panels making comparison difficult. for example is D1 in panel A equivalent to 24 hours in panel C, if so there is a discord between the gene expression and protein levels that should be explained.

Figure 2. There is a broad range in the degree of responses to ZIKV infection in different tissues. Does this reflect differential expression levels of c-FLIP (or caspase 3 or 8) in different tissues? Only western blot data is shown for spleen in fig S2, data for other organs should also be include

---

## [Decision Letter · Decision Letter 1]

28 Jun 2024

Dear Mis Luo,

Thank you very much for submitting your manuscript "c-FLIP facilitates ZIKV infection by mediating caspase-8/3-dependent apoptosis" for consideration at PLOS Pathogens. As with all papers reviewed by the journal, your manuscript was reviewed by members of the editorial board and by several independent reviewers. The reviewers appreciated the attention to an important topic. Based on the reviews, we are likely to accept this manuscript for publication, providing that you modify the manuscript according to the review recommendations.

All of the reviewers agreed that the manuscript is much improved and acknowledged the significant amount of effort that went into revising the manuscript. Reviewer 2 raised a few very minor issues that still remain, which I agree need addressing. These amendments do not require any additional experiments but the suggested edits should be made before we can consider this manuscript for publication.

Sincerely,

Kevin Maringer, PhD

Guest Editor

PLOS Pathogens

Michael Letko

Section Editor

PLOS Pathogens

Michael Malim

Editor-in-Chief

PLOS Pathogens

orcid.org/0000-0002-7699-2064

All of the reviewers agreed that the manuscript is much improved and acknowledged the significant amount of effort that went into revising the manuscript. Reviewer 2 raised a few very minor issues that still remain, which I agree need addressing. These amendments do not require any additional experiments but the suggested edits should be made before we can consider this manuscript for publication.

Reviewer Comments (if any, and for reference):

Reviewer's Responses to Questions

**Part I - Summary**

Reviewer #1: The authors have done a good job of answering queries and suggestions. Some points were raised by multiple reviewers and extra experiments, repeats of experiments and reanalysis of data are presented.

Reviewer #2: The authors have adequately addressed all the previous concerns, and this reviewer appreciates the updated version of this manuscript, which is much more compelling in assessing the role of c-FLIP in Zika virus infection.

Reviewer #3: I appreciate the authors efforts to address my comments. The additional data and text edits provided have improved the clarity of the manuscript.

**Part II – Major Issues: Key Experiments Required for Acceptance**

Reviewer #1: (No Response)

Reviewer #2: (No Response)

Reviewer #3: No additional experiments suggested

**Part III – Minor Issues: Editorial and Data Presentation Modifications**

Reviewer #1: (No Response)

Reviewer #2: Only very minor comments still need to be considered:

1. Line 146: What does c-FLIPS replication mean in this context?

2. Fig 1D: c-FLIPL is missing in the y-axis.

3. Indicate the titres plot in Fig 1F and discuss their fold-reduction.

4. According to Figure 2, lines 227-228 need to be rephrased as they are not true.

5. Line 333: remove the abbreviation of “it’s”

6. Although differences are significant in Figs 6A and 6B, it would be interesting to discuss further the fold-change differences, which, according to the graph, are minimal.

7. Apart from cleaved caspase 3, in Fig 6D, it is very difficult to assess caspase activation without proper quantification. Full-size blots in the Supplementary Information will be required to determine this.

8. Fig 7D is very difficult to assess. Images with better contrast will be required to show annexin V-FITC.

Reviewer #3: No modifications suggested

PLOS authors have the option to publish the peer review history of their article (what does this mean?). If published, this will include your full peer review and any attached files.

Reviewer #1: No

Reviewer #2: No

Reviewer #3: No

Figure Files:

Data Requirements:

Reproducibility:

References:

---

## [Editor Report · Decision Letter 2]

9 Jul 2024

Dear Mis Luo,

We are pleased to inform you that your manuscript 'c-FLIP facilitates ZIKV infection by mediating caspase-8/3-dependent apoptosis' has been provisionally accepted for publication in PLOS Pathogens.

Best regards,

Kevin Maringer, PhD

Guest Editor

PLOS Pathogens

Michael Letko

Section Editor

PLOS Pathogens

Michael Malim

Editor-in-Chief

PLOS Pathogens

orcid.org/0000-0002-7699-2064
---

## [Editor Report · Acceptance letter]

18 Jul 2024

Dear Mis Luo,

We are delighted to inform you that your manuscript, "c-FLIP facilitates ZIKV infection by mediating caspase-8/3-dependent apoptosis," has been formally accepted for publication in PLOS Pathogens.

Best regards,

Michael Malim

Editor-in-Chief

PLOS Pathogens

orcid.org/0000-0002-7699-2064